# Strengthening of Concrete Column by Using the Wrapper Layer of Fibre Reinforced Concrete

**DOI:** 10.3390/ma13235432

**Published:** 2020-11-28

**Authors:** Peter Koteš, Martin Vavruš, Jozef Jošt, Jozef Prokop

**Affiliations:** 1Department of Structures and Bridges, Faculty of Civil Engineering, University of Žilina, Univerzitná 8215/1, 010-26 Žilina, Slovakia; martin.vavrus@uniza.sk; 2Laboratory of Civil Engineering, University of Žilina, Univerzitná 8215/1, 010-26 Žilina, Slovakia; jozef.jost@uniza.sk (J.J.); jozef.prokop@uniza.sk (J.P.)

**Keywords:** member resistance, fibre reinforced concrete, carbon steel, shear resistance, column, strengthening

## Abstract

Structures and bridges are being designed on the proposed and requested design lifetime of 50 to 100 years. In practice, one can see that the real lifetime of structures and bridges is shorter in many cases, in some special cases extremely shorter. The reasons for the lifetime shortening can be increased of the load cases (e.g., due to traffic on bridges, or due to other uses of a structure), using the material of lower quality, implementation of new standards and codes according to Eurocode replacing older ones. During the whole lifetime the structures must be maintained to fulfil the code requests. If the constructions are not able to fulfil the Ultimate Limit States (ULS) and the Serviceability Limit State (SLS), the structures or bridges have to be strengthened (whole or its elements). The purpose of the paper is the presentation of using a layer of the fibre concrete for a columns’ strengthening. Using the fibre reinforced concrete (FRC) of higher tensile strength makes it possible to increase the load-bearing capacity of the cross-section the column. The contact between the old concrete (core of column) and newly added layer (around column) is very important for using that method of strengthening. In the article, there is also a comparison of the surface modification methods.

## 1. Introduction

Fibre reinforced concrete (FRC), or concrete with dispersed reinforcement, is a rheological composite material based on silicates with ideally dispersed reinforcement elements in the form of fibres—either steel or synthetic. The dispersion reinforcement significantly affects some shortcomings of conventional concrete [1,2,3]. The advantage of fibres is best visible in spatial mechanical stress, where they absorb the tensile forces in areas of cement putty and reduce the brittle nature of the concrete failure, where the flexural tensile strength, transverse strength or simple tensile strength increases for about 30–50%, as compared to plain concrete properties, according to [4,5]. The fibre concrete can also better withstand the volume changes due to shrinkage during the solidification and hardening and due to changes in ambient temperatures [6]. Compared to plain concrete, fibre concrete is more valuable and tougher. In the case of the steel fibres, there can be a problem with a corrosion of fibres and reinforcement inside the cross-section [7,8,9] and following crack pattern [10,11], which can cause a decrease in the cross-section resistance [12].

Fibre reinforced concrete belongs to High Performance Concretes (HPC), which are typically used for strengthening linear horizontal members, for instance, beams [13,14]. The mechanical properties of fibre reinforced concretes are influenced by the type of used fibres, such as steel, glass, polypropylene of basalt [15,16,17]. The research [18] is focused on the mechanical properties and chloride resistance of concrete reinforced with hybrid polypropylene and basalt fibres. Some researches are also focused on using fibre reinforced concrete for column strengthening [19], as in this paper, but without taking into account the influence of contact between old and new concrete and the columns are only in compression. The originality of the article is the use of FRC to strengthen vertical elements, such as columns, stressed by a combination of compression and bending (using the good properties of FRC not only in compression but also in tension), taking into account the contact resistance between the old and new parts of the cross-section.

Results from the 3-point (Figure 1a) and 4-point bending test or the wedge split test (Figure 1b) [20,21,22] are important for the fibre concrete. From those tests, the necessary input data for estimating the strength of fibre concrete are obtained.

The softening or hardening of the fibre reinforced concrete occurs after the appearance of the first crack in the composite. The crack occurs in the brittle concrete matrix, the load is carried by the fibres. If the collapse of the element is to be prevented, the load-bearing capacity of the fibres $\sigma_{fu}V_{f}$ should be greater than the load applied to the element. This relationship can be classified based on the elastic stress during the crack formation in the cement composite [5,23]
(1)$$\sigma_{fu}\;V_{f}>E_{m}\;\varepsilon_{mu}V_{m}+E_{f}\;\varepsilon_{mu}V_{f}$$
where *E_m_* is the modulus of elasticity of concrete, *E_f_* is the modulus of elasticity of the fibre reinforced concrete, ε_mu_ is the cracking strain of matrix in composite, V*_m_* is the volume of concrete matrix, *V_f_*
*σ_fu_* is the load-bearing capacity of fibres. 

If Equation (1) was satisfied (i.e., when the fibre content $V_{f}$ is sufficient), the first crack that occurs in the fibre reinforced concrete composite would not lead to a catastrophic failure, but it would cause a redistribution of the load between the matrix and the fibres. This means that the load transmitted by the matrix in the crack zone is acting on the fibres that bridge the crack and the cement matrix is stressed at the edges of the crack. The additional load would result in the development of further cracks, until the matrix is divided into several segments separated by cracks [24,25].

According to the FIB model code [20], determination of the residual flexural tensile strength is performed. The test result represents the load-CMOD dependence. The test should be in accordance with EN 14651 [26]. When performing the test, the value of the load $F_{Ri}$ is measured at certain degrees of crack opening $CMOD_{i}$. Subsequently, the residual strength is determined either according to Equation (2) or directly from the course of the test (Figure 2)
(2)$$f_{R}=\frac{3\;F\;l}{2\;b\;h_{sp}^{2}}$$

Denotation of the input data, given in Equation (2), is shown in Figure 1. Values of $f_{R1}$. and $f_{R3}$ correspond to values of $CMOD_{1}=0.5\;\mathrm{mm}$ and $CMOD_{3}=2.5\;\mathrm{mm}$, respectively (Figure 2).

Residual strength limits must be determined for the input parameters, for the residual serviceability of tensile strength $f_{Fts}$ and for the final residual ultimate tensile strength $f_{Ftu}$. The residual ultimate strength is calculated according to Equation (3) and the residual strength according to Equation (4)
(3)$$f_{Fts}=0.45.f_{R1},$$
(4)$$f_{Ftu}=f_{Fts}-\frac{w_{u}}{CMOD_{3}}\left(f_{Fts}-0.5f_{R3}+0.2f_{R1}\right),$$
where *w_u_* is the ultimate crack opening acceptable in the structural design, *f_Fts_* is the serviceability residual strength, *f_Ftu_* is the ultimate residual strength. 

Calculation of the resistance moment, according to the FIB MODEL CODE, is based on the balance of internal forces. Figure 3 shows a simplification of the stress distribution along the cross-section height, where the linear stress distribution after the crack formation (a) and the rigid plastic stress distribution (b) are considered.

The second type of the residual strength calculation of the fibre reinforced concrete is conducted according to the RILEM TC 162—TDF [27] (see Figure 4). That calculation stems from establishment of the load values at certain CMOD crack opening widths.

From the measured values of the crack width and load, it is necessary to determine the stresses according to the following Equations (5)–(7)
(5)$$\sigma_{1}=0.7f_{fctm,fl}\left(1.6-d\right),$$
(6)$$\sigma_{2}=0.45f_{R1}k_{h},$$
(7)$$\sigma_{3}=0.37f_{R4}k_{h}.$$

The paper presents the possibility of using fibre reinforced concrete to strengthen columns taking into account the effect of contact between old concrete and new fibre reinforced concrete on the load-carrying capacity of the cross-section. The practical use of the work can be to strengthen the columns of frame structures or piers of the substructure of bridges with insufficient load-carrying capacity of the cross-section [28,29,30]. It is possible to solve the mentioned problems also using probabilistic analysis [31,32,33,34]. The problem, which is not solved in the paper, is also resistance of fibre reinforced concrete against chloride ion penetration and diffusion coefficients [35,36]. 

## 2. Experimental Programme

Most experiments for column strengthening deal with the compressive force and low bending moment only. In this case, the most suitable method is strengthening by the concrete encasement with a layer of reinforced concrete, plain concrete and wrapping with the FRP sheets [37,38,39]. It this case, the slippage between the layers is not assumed—rigid contact is assumed. The members stressed by a combination of normal force and bigger bending moment are no longer advantageously reinforced by these methods, in which case the methods of reinforcement using steel elements (L-profiles) or the use of FRP lamellas are suitable, which are more technologically demanding to ensure cohesion and anchorage [40]. Therefore, the purpose of this research was to verify the use of modern material such as fibre reinforced concrete. By using this material and the method of strengthening, an attempt is made to find a compromise between the increased load-bearing capacity and cross-section enlargement to a lesser extent than in the case of reinforced concrete wrapping (minimal cross-section enlargement is by using FRP sheets or steel plates [41,42]). With such stressed elements, there are significant shear stresses between the surface of the old core and the new layer. For this reason, various methods of surface treatment [43] (contact between the original core and the new reinforcement layer) were investigated using the so-called Push tests.

### 2.1. Analysis of the Shear Connection between the Old Concrete and the New Fibre Reinforced Concrete

The load-bearing capacity of the strengthened member (the whole cross-section) depends in part on the shear resistance between the old concrete (core) and a layer of the newly added concrete [44,45,46]. This only applies if sliding between the layers is allowed. This does not apply if slipping is prevented, for example, due to the limitation of the lateral dilatation of the inner core closed by the concreted layer. Shear tests were performed to determine the shear resistance. Implementation of tests should prove the relevance of the coefficients of cohesion *c* and friction *μ* for individual types of surfaces. The bearing capacity for transmission of the shear force along the contact surface was also verified for the individual surface treatment methods, which were used for the following experiments [47].

Three methods of contact treatment (smooth surface, indent and notches) with different types of reinforcement in the case of smooth surfaces were used for surface treatment (Figure 5):(A)Smooth surface wrapped with fibre reinforced concrete (fibre reinforced concrete with steel wires)—(VH),(B)Indent wrapped with fibre reinforced concrete—(ZZ),(C)Notches wrapped with fibre reinforced concrete—(ZR),(D)Smooth surface wrapped with reinforced concrete—(ZH),(E)Smooth surface wrapped with plain concrete—(PH),(F)Smooth surface wrapped with foil and a layer of fibre reinforced concrete—(FH).

Selection of the surface treatment was performed with regard to the complexity of design in practice. Consideration was also given to the shear strength of individual surface modifications.

Dimensions of the column were chosen with the cross-section dimensions of 0.16 m × 0.16 m and a height of 0.35 m. The column (core) was reinforced with 4Ø10 mm, the length of the bars is 0.3 m. The stirrups of diameter 8 mm were placed at an axial distance of 85 mm (Figure 6). Reinforcement of type B 500B was used. The concrete cover of reinforcement was based on the XC1 class of environment and a design life of 50 years. The concrete cover of the main reinforcement was determined at 20 mm. The concrete class C 16/20 was used. Thickness of encasement (adding layer) was set at 35 mm. 

Samples, after concreting, were treated for the next three days to prevent formation of the shrinkage cracks and after 28 days the samples were removed from the forms. Subsequently, the samples´ surfaces were modified according to the above-mentioned surface treatments (Figure 7). All the samples were free of impurities and dust particles. The sample wrapped with the foil (FH) was made only due to the separation of the cohesion and friction. Subsequently, the effect of the wrapping on the load was observed.

Temporary polystyrene adhered to the bottom part of the treated samples. The samples thus prepared were placed in the formwork with centring the spacer screws (Figure 7d). Prior to concreting, the samples were moistened and then the samples were poured—concreted with fibre reinforced concrete or concrete. For the concrete mixture without fibres, the same recipe with wire concrete was used, only the fibres were not used. The composition of the concrete mix was based on previous research [3], where the amount of fibres added to the concrete mix and their effect on the flexural tensile strength were compared (Table 1). Design of the concrete mixture was also based on the experience of the implementation team in order to design the concrete that can be produced in real conditions (i.e., in normal production and not only in laboratories) and where it is possible to easily dose up to 60 kg of fibres per 1 m^3^ of concrete, Table 1.

The weight of the fibres Dramix 3D equal to 40 kg/m^3^ represents a fibre ratio of 0.5%. During the concreting, test cubes with dimensions of 150 × 150 × 150 mm^3^ were cast in order to determine the strength characteristics of concrete. The compressive strength test of the concrete was conducted on the day of the “push tests”. Samples were concreted in two batches (A, B), 3 test samples were cast from each batch. Their cube strengths *f_c,cube_* are given in Table 2.

### 2.2. Load Set for the “Push Test”

The samples were placed in a jack, a steel plate was placed on the upper edge of the sample for better load distribution and then the sensors for distance measurement No. 1 and No. 2 were glued to the samples (Figure 8). The displacement between the core and the encasement was measured. Loading was executed with a press with a maximum force of 1000 kN. The loading rate was 0.5 mm/min.

For comparison, the characteristic values of the shear strength *τ* [48] were calculated according to Equation (8) using Formula (9), the values of which were 104.96 kN for the smooth surface and the shear strength was calculated at 151.04 kN for indent
(8)$$\tau_{n}=c+\mu \sigma_{n},\;(\mathrm{kN}\cdot m^{-2})$$
(9)$$F_{failure}=\tau_{n}\cdot S,\;(\mathrm{kN})$$
where *c* is cohesion (kN m^−2^), *μ* is the coefficient of friction (-), *σ_n_* is the normal stress (kN m^−2^), *τ_n_* is the shear stress (kN m^−2^), *S* is the area of contact shear surface (m^2^). 

The standard values of the coefficient of friction *c* = 0.20 for a smooth surface, *c* = 0.40 for a surface with notches and *c* = 0.50 for a surface with indent were used to calculate the maximum failure force *F_failure_* according to the standard [49]. In this case, the normal stresses *σ_n_* is considered to be zero (*σ_n_* = 0.0 kN m^2^). Then the coefficient of friction μ does not need to be considered. The calculated values are shown in Table 3 and they are compared to values achieved from the experiment.

### 2.3. Results of the “Push Test” Measurements

Sample tests were prepared 55 days after concreting. The main goal was to determine the shear resistance for individual methods of the surface treatment (processing). Seventeen samples were tested and results are shown in Figure 9. The slip value between the core and the concreting layer is the average from the two sensors.

From the obtained results (Table 3), the calculations from Equations (8) and (9) were confirmed, where the values of the tests in some cases exceeded the calculated results for the smooth surface and notch many times (Table 4). As the standard [49] does not specify the coefficients of friction and cohesion for other surface treatments, it was necessary to calculate those values by reverse-calculation and compare them to the standard values or underestimation of the values recommended in the standard. However, it should be noted that the results came from large differences between the individual samples, except for the sample concreted with plain concrete, where there was a sudden breakage of the concreting layer. The variety of results is largely due to the concrete mix and air cavities at the contact surfaces of the encasement.

### 2.4. Experimental Verification of the Strengthened Columns

For experimental verification of the strengthened columns, 3 sets (groups) of columns were made for two samples, i.e., a total of 6 experimental samples of columns. The first group consisted of non-strengthened columns (2 specimens), which were used to determine the basic value of load-bearing capacity. The second group of columns (2 specimens) was strengthened with a layer of fibre reinforced concrete (concreting with a layer of fibre reinforced concrete with steel wires) and the last group (2 specimens) was reinforced with the new layer of reinforced concrete.

#### Geometry and Reinforcement of Experimental Non-Reinforced Columns

The basic dimensions of the column were determined to be 160 mm × 160 mm and the height 2500 mm, which were based on the experiments in [49,50]. 

The lower class C16/20 concrete was used. Steel B 500B was used to reinforce the column. The column was reinforced with 4Ø10 mm and stirrups of a diameter of 8 mm at an axial distance of 100 mm (Figure 10). It was assumed that the minimum and maximum degree of reinforcement *ρ_min_* = 0.002 ≤ *ρ* = 0.0123 *≤ ρ_max_* = 0.04 was observed. The slenderness of the column is equal to the value *λ* = 54.13 > *λ_lim_* = 21.95, so the column is considered as slender. The environmental XC1 class was considered. Steel plates were attached to both ends of the columns and mounted before concreting. The samples were treated for 3 days. 

According to [49] and the cross-section, an interaction diagram was constructed. As mentioned in the previous section, the experiment was focused on the combination of the axial pressure and bending moment. The point with the largest bending moment was selected from the diagram. At that point, the column is able to transmit a maximum characteristic bending moment of 16.26 kNm at a normal compressive force of 157.79 kN. Based on these values, the eccentricity was determined to be equal to *e* = 100 mm.

### 2.5. Columns Strengthening 

Two strengthening methods were considered in the experimental program. The first alternative was the fibre reinforced concrete reinforcement (steel fibre reinforced concrete concreting—denotation of samples SV). The thickness of the layer was considered to be 35 mm and was chosen with regard to slenderness. The new cross-section dimensions were increased to 230 mm × 230 mm, where the slenderness decreased to *λ* = 37.65 > *λ_lim_* = 22.94, but the strengthened columns still remained as slender. The chosen thickness of the new layer also took into account the minimum coverage of the main longitudinal reinforcement in the new layer of concrete in the second method of reinforcement [51].

A concrete mix of fibre reinforced concrete, identical to the “push tests”, was used for the cladding (Figure 5). At the same time, concrete compressive strength cubes and beams were removed to determine the residual flexural tensile strength.

The second alternative (method) was to strengthen the column with a layer of reinforced concrete of the same thickness as in the first case, namely 35 mm (layer of concrete with reinforcement—denotation of samples SB). The cross-section was additionally reinforced with longitudinal reinforcement 4Ø10 mm (in the newly added layer) and stirrups with a diameter of 6 mm at axial distances of 150 mm (Figure 11). In this case, the minimum thickness of the cover layer for the stirrups was not observed due to the comparison of the cross-sections (thickness of the reinforcing layer). If one wanted to adhere to the cover layer for stirrups in the XC1 environment, the minimum layer thickness would have to be 46 mm (if one considers the main additional reinforcement Ø10 mm, stirrups Ø6 mm and the distance of the main reinforcement from the original concrete at least 10 mm). The concrete mix was identical to the recipe as in the first alternative, only without the addition of fibres.

### 2.6. Loading Assembly for Columns 

Within the laboratory conditions and due to the column height (laboratory options), the test itself was performed horizontally (in a horizontal position). The assembly consisted of transverse girders and two Dywidag longitudinal bars. A press with a maximum force of 3000 kN was placed on the transverse girders. The columns were supported at both ends of the joints. The measuring devices S1–S5 were arranged along the entire length (column height), (Figure 12). These sensors measured the horizontal deformation of the column. The sensor S6 measured the longitudinal deformation from which the rotation of the end section of the column could be deduced. String strain gauges T1 and T2 were installed in the middle of the column span, which were used to measure the tension in the end fibres.

### 2.7. Results of Experimental Measurements 

All the members were loaded during the test with an axial force, the increment of the force was 25.0 kN until the deformation stabilised at a particular level. Six samples were tested. Results of the maximum forces during the columns´ failure are shown in Table 5.

### 2.8. Comparison of Column Strengthening Methods

It is clear from the results of the experiment that a more effective method of strengthening is reinforcement with the new reinforced concrete layer in terms of a higher increase in resistance than has already been stated (Figure 13). This strengthening increased the load-bearing capacity of the column on average value 557.2 kN, which is about 172.2 kN more than the strengthening with a layer of fibre reinforced concrete. The shape of the load-deformation curves (Figure 13) is not completely identical for the individual methods of reinforcement, which can be caused, for example, in the case of fibre-reinforced concrete, due to the partially uneven distribution of fibres, or in the case of reinforced concrete reinforcement, there could be a slight shift in the additional reinforcement bars during the concreting.

However, by comparing the load-bearing capacity of the cross-section using these two different methods of strengthening, using the material consumption (concrete reinforcement versus fibres), it is obtained that the use of fibre is more efficient. The total reinforcement consumed (longitudinal and stirrups) for the strengthening concrete reinforcement was 12.37 kg per column, but the weight of the fibres for the strengthening concrete reinforcement was 3.05 kg per column, which is four times less weight compared to strengthening concrete reinforcement (Figure 14). Taking into account the material consumption, for increase of the load-bearing capacity, it is obtained that 0.030 kg of reinforcement had to be used to increase the load-bearing capacity of the cross-section by 1 kN, but in the case of the fibre reinforced concrete, it was only about 0.013 kg of fibres, which is approximately 2.5 times less. The above information is valid to our selected fibre ratio of 0.5% up to 1.5%. Increasing the fibre ratio over 1.5% to 2.0% does not effectively increase the cross-section resistance. 

### 2.9. The Fibre Reinforced Concrete

In order to obtain the necessary material properties of the fibre reinforced concrete, a test was performed to determine the flexural tensile strength of concrete. The test was performed with a 3-point bending test. Three beams of dimensions 150 × 150 × 700 mm^3^ with a notch were tested. The height of the notch corresponded to 1/6 of the height of the beam. The test was performed according to the standard [52,53]. The sample was loaded at a constant rate of 0.05 mm/min until the crack opening CMOD = 0.1 mm, then the loading rate was increased to 0.2 mm/min. The end of the test occurs when the crack opening value CMOD = 4 mm is reached. The resulting values are shown in Figure 15.

## 3. Numerical Analysis

To verify the samples from the experimental program, the 3D models were created in the nonlinear program ATENA 3D Červeka Consulting [54]. The program is based on the deformation method of finite elements and the main characteristics of the application of the nonlinear model materials. This makes it possible to analyse building structures or their parts in critical conditions where there are failures.

### 3.1. Numerical Analysis of the Fibre Reinforced Concrete

The input values of materials for the fibre reinforced concrete were not easy to determine. For example, the fracture energy cannot be determined in experiments (experiments cannot be brought to a state where it would be possible to measure the total value of the fracture energy) [54,55]. Therefore, the load-bearing capacity of beams from 3- and 4-point bending tests was simulated in the program ATENA and the material parameters are set so that the response corresponds as much as possible to the actual beam [56,57]. Used input values are shown in Table 6.

### 3.2. Numerical Analysis of the “Push Tests”

Numerical models have been developed to verify the “push tests” (tests to verify the contact between the old part of the column and the new layer). The 3D models consisted of a core, encasement and a steel plate (Figure 16). The entire structural element was created from 3D spatial elements. To determine the shear resistance, it was necessary to adjust the contacts of the 3D interface for each surface separately.

### 3.3. Numerical Contact Between the Fibre Reinforced Concrete with a Smooth Surface (VH)

The numerical model was adjusted (calibrated) to obtain an average value from the measured experimental values. The results of the numerical analysis (VH_A) indicate that the shear strength was reached at a force of 141.756 kN (Figure 17).

The setting of the contact is enabled by parameters K_nn_ and K_tt_, which indicate normal and shear (tangential) stiffnesses, respectively [54]. The resulting properties of the interface used in the numerical model are shown in Table 7. 

### 3.4. Numerical Analysis of a Non-Strengthened Column

Numerical 3D models were also made to verify experimental column samples. The non-strengthening model consisted of a column, vertical and horizontal reinforcement and distribution plates (Figure 18a–c). The “Brick” network element type was used for macro elements, with a relative coefficient of 0.05. The macro element consisted of 5760 elements (Figure 18d).

Material characteristics for individual elements are given in Table 8. The model was loaded by a controlled deformation at an eccentricity of 100 mm with a value of 1.0 mm in all the loading steps. The Arc-length method was used for the calculation type. Monitors were placed on the model to the same points as in the experiment.

The resulting model was adapted to the resulting average values from two experimental samples. Deformations at individual locations are shown in Figure 19.

### 3.5. Numerical Analysis of the Reinforced Column Strengthened 

For modelling the column with strengthening, the 3D model of the non-strengthened column from the previous section was used. This means that the model was supplemented with a 35 mm thick layer (Figure 20c) with material properties determined from numerical analysis of the fibre-reinforced concrete. The GAP contact was also inserted between the individual surfaces (old core of column and new layer of fibre reinforced concrete). The contact parameters were modelled as in Section 3.3.

The resulting values from the path sensors are shown in Figure 21 and Figure 22.

## 4. Conclusions

The results of experimental measurements and numerical modelling have demonstrated the suitability of using new types of materials, such as FRC, for the strengthening of vertical elements, such as columns.

Based on results of the experimental program and numerical analysis, it is clear that the surface (contact) treatment on push-test samples results in an increase in the shear resistance. However, in comparison to the experimental values, values calculated according to the standard [49], in some cases, were several times higher. This phenomenon is caused by the concrete mixture, compaction of fresh concrete and subsequent removal of air pores. The overgrowth of the original and new concrete, which occurred during the hydration of the cement putty, significantly affects the resistance of the contact. When the specimens failed, one or two major longitudinal cracks were created, which occurred at the moment of reaching the shear resistance. After reaching the maximum shear force, the crack was opened and the sample lost shear resistance. In the case of fibre concrete with a smooth surface (VH), the fibres were activated after the tensile strength of the concrete was reached and began to transmit tensile stresses caused by the pushing of the core into the wrapping. This phenomenon was mostly cleared up in the samples with the indent, where the indent was pushed into the casing and tensile stresses along the cross-section of the casing were raised. In the case of plain concrete (PH) samples, the strengthening cover was damage and after the formation of a crack, a part of the strengthening cover fell off. The use of concrete reinforcement did not lead to significant cracking, the strengthening cover had sufficient tensile strength, so there was only shear failure between the individual layers of surfaces.

Two alternatives were considered for strengthening in the experimental program of column strengthening. The first alternative was strengthening using a layer of fibre concrete. It was a layer 35 mm thick. After the failure of the specimen, the formation of one major crack at the point of greatest stress was seen. Due to the formation of one major crack, it can be established that the composite has been softening. After closer analysis, it was clear that the fibres from the cement matrix were pulled out. Overgrowth of new and original concrete was also observed, during which the individual parts (core and encasement) did not slip. It follows that the strengthening column with a fibre volume of 0.5% carried out 258.19% (235.925 kN) more than the unreinforced column. However, in comparison with the strengthening by reinforcement concrete, there was a brittle failure in the pressure area. Even in this case, there was no shear displacement in the contact. Compared to fibre-reinforced concrete, it carries out an average of 144.71% (172.18 kN) more.

## Figures and Tables

**Figure 1 materials-13-05432-f001:**
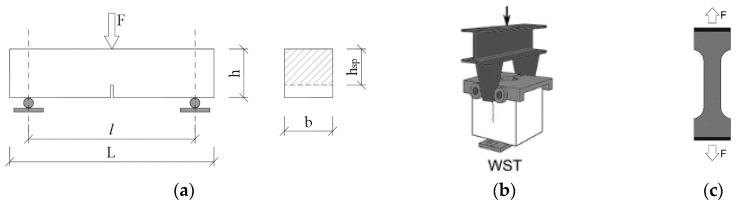
(**a**) 3-point bending test, (**b**) wedge splitting test, (**c**) direct tension test (dog bone).

**Figure 2 materials-13-05432-f002:**
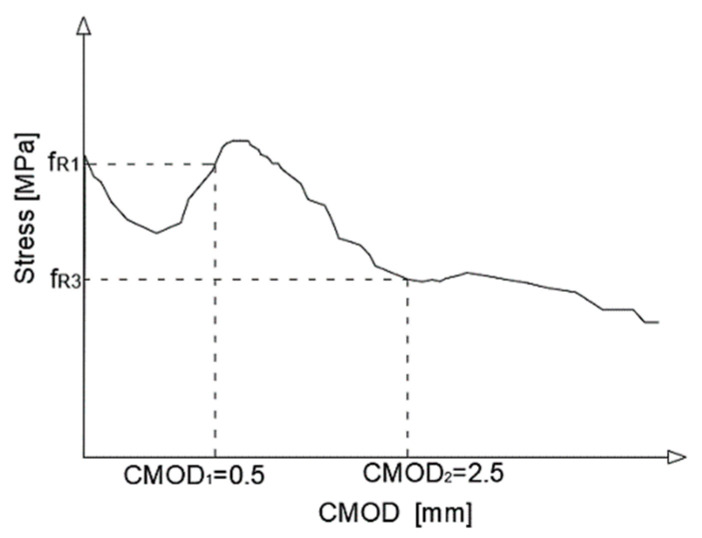
Course of the 3-point bending test.

**Figure 3 materials-13-05432-f003:**
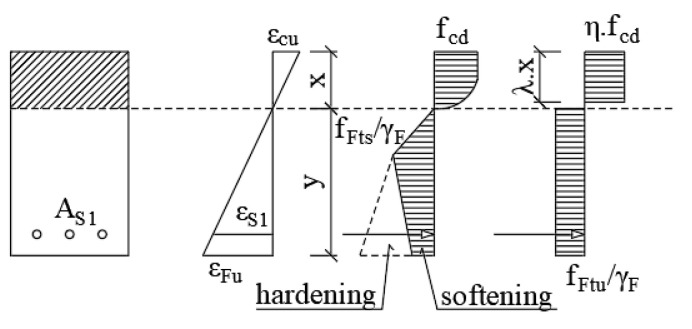
Simplification of the cross-section stresses.

**Figure 4 materials-13-05432-f004:**
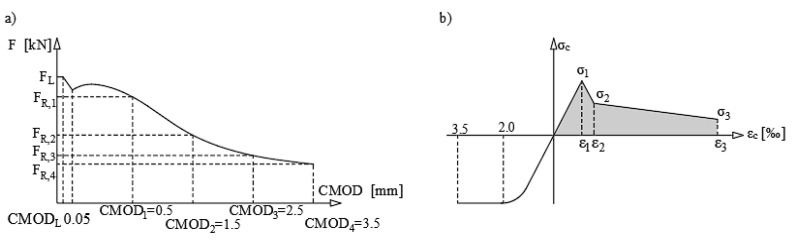
(**a**) Course of the 3-point bending test, (**b**) simplification of the cross-section stresses according to the RILEM TC 162–TDF [27].

**Figure 5 materials-13-05432-f005:**
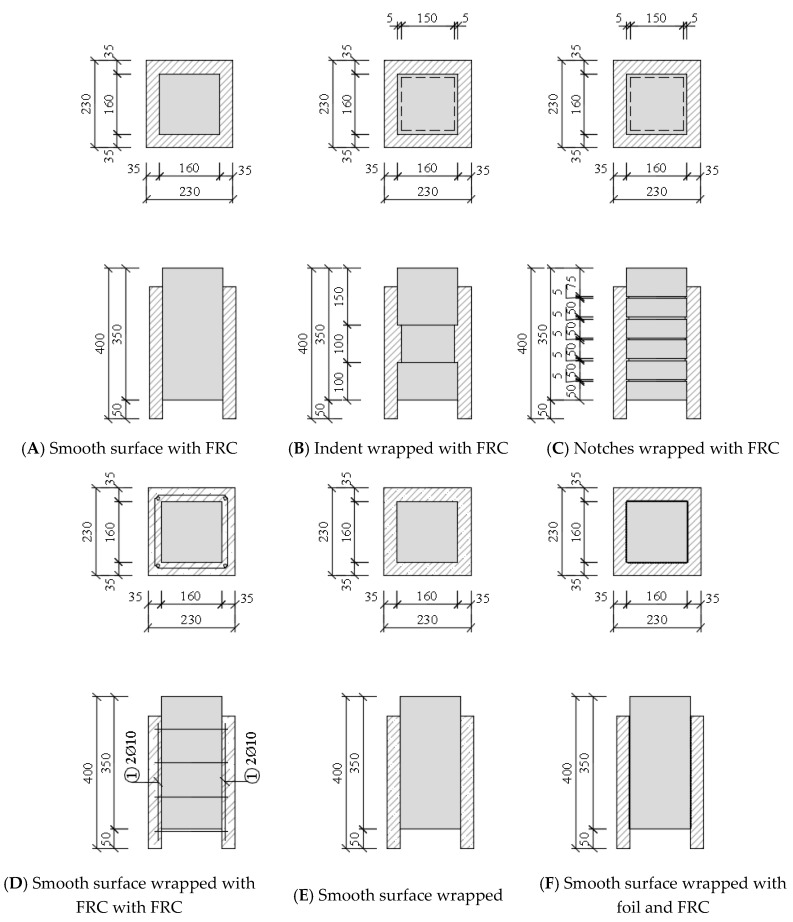
Different methods of surfaces modification.

**Figure 6 materials-13-05432-f006:**
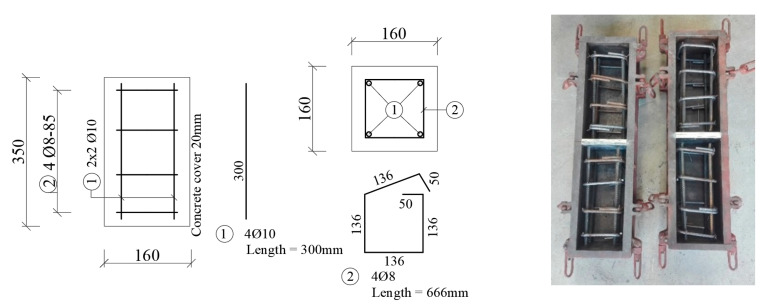
Shape and reinforcement of the column (core) for the Push-test before concreting.

**Figure 7 materials-13-05432-f007:**
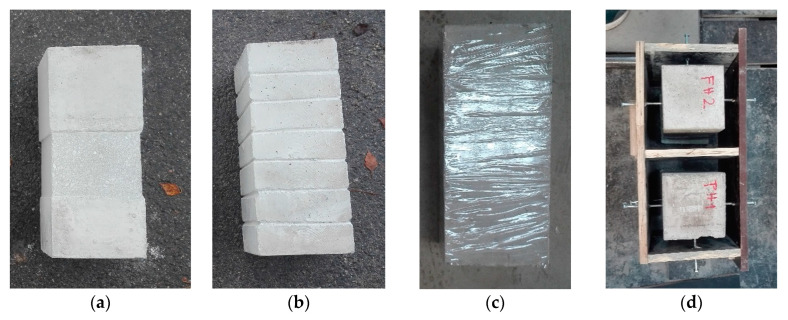
Modified column for was observed Push-test, (**a**) indent, (**b**) notches, (**c**) wrapped surface with foil, (**d**) formwork with centring spacer screws.

**Figure 8 materials-13-05432-f008:**
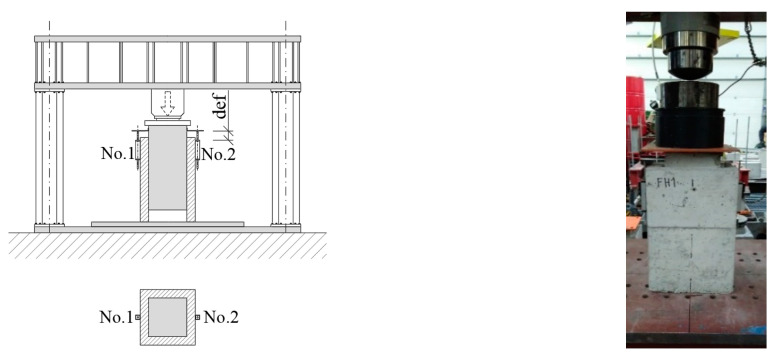
Load assembly.

**Figure 9 materials-13-05432-f009:**
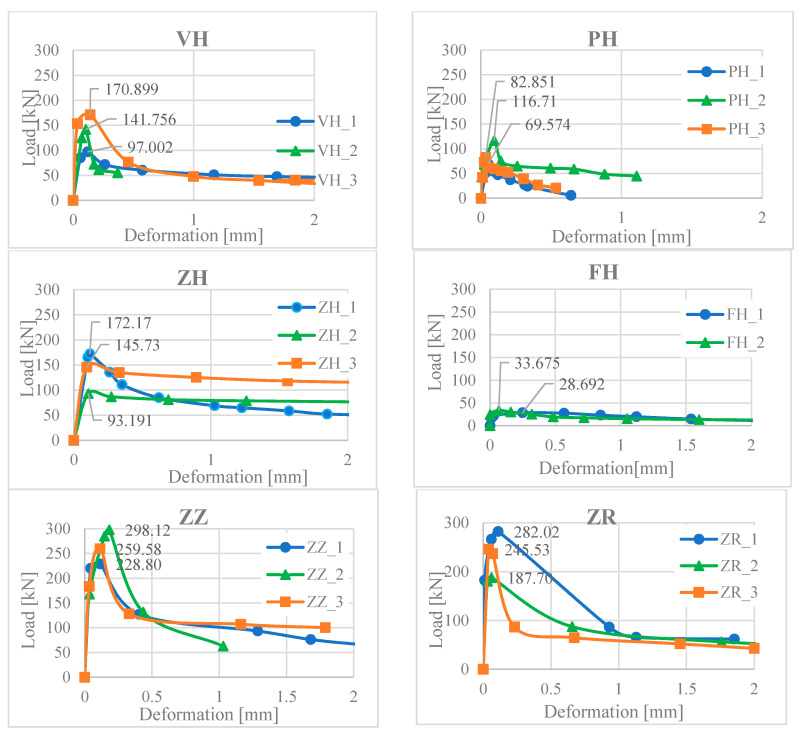
Loading force—deformation for different surface modifications.

**Figure 10 materials-13-05432-f010:**
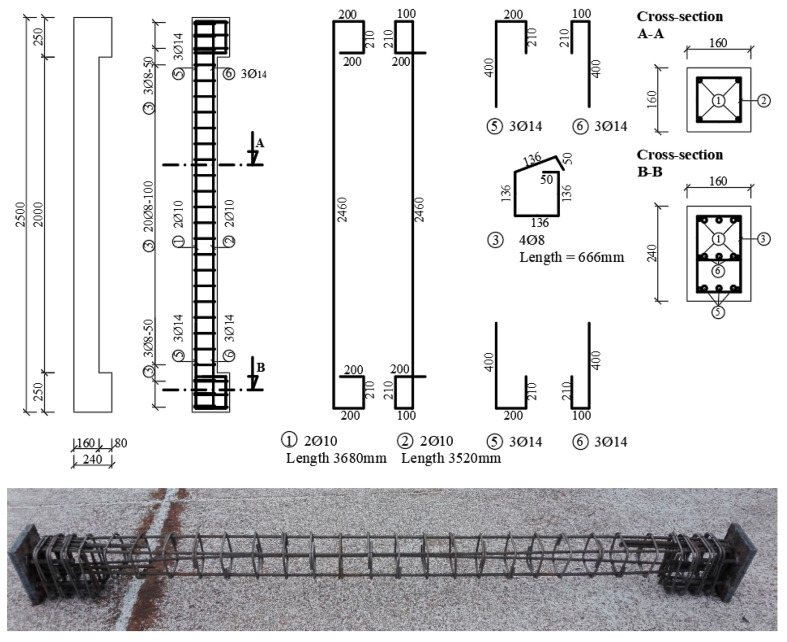
Reinforcement of the column.

**Figure 11 materials-13-05432-f011:**
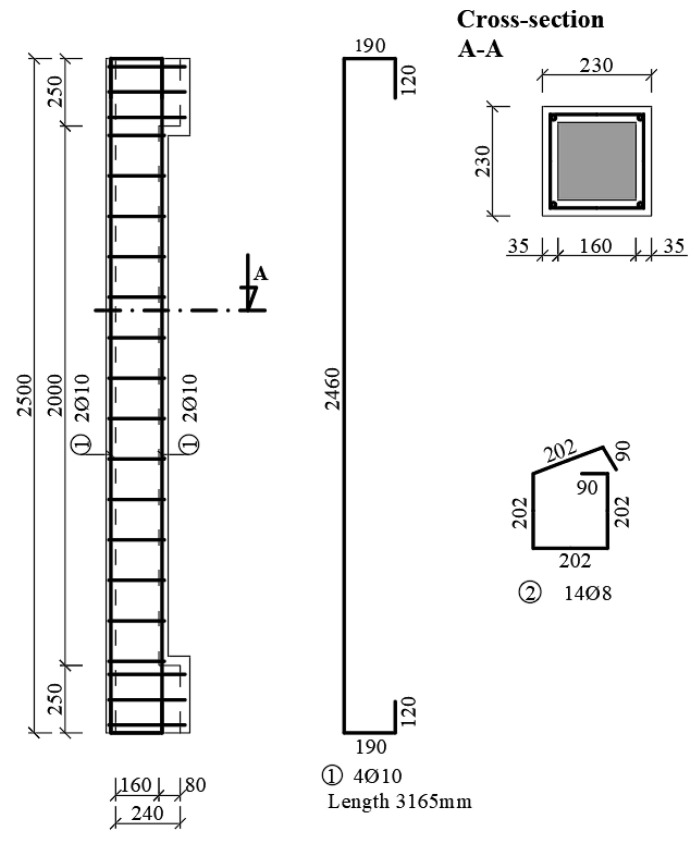
Strengthening of column. A_core_ = 0.16 × 0.16 = 0.0256 m^2^ A_total_ = 0.23 × 0.23 = 0.0529 m^2^ A_FRC_ = A_total_ − A_core_ = 0.0273 m^2^.

**Figure 12 materials-13-05432-f012:**
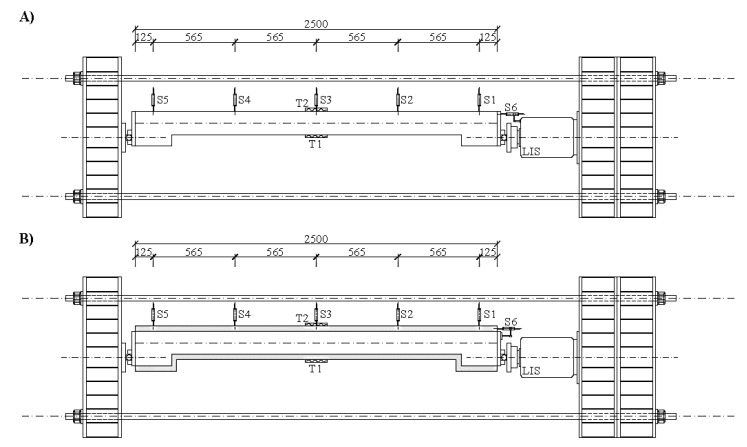
(**A**) Loading assembly for the non-strengthened columns, (**B**) Loading assembly for strengthened columns. Unit: mm.

**Figure 13 materials-13-05432-f013:**
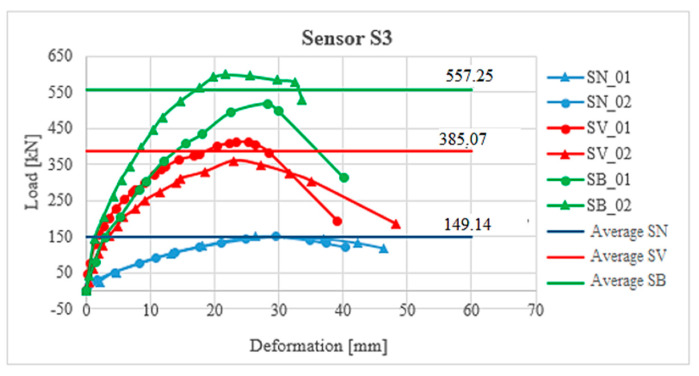
Comparison of strengthening methods—sensor S3 (sensor S3 was in the middle of the column length, see Figure 12).

**Figure 14 materials-13-05432-f014:**
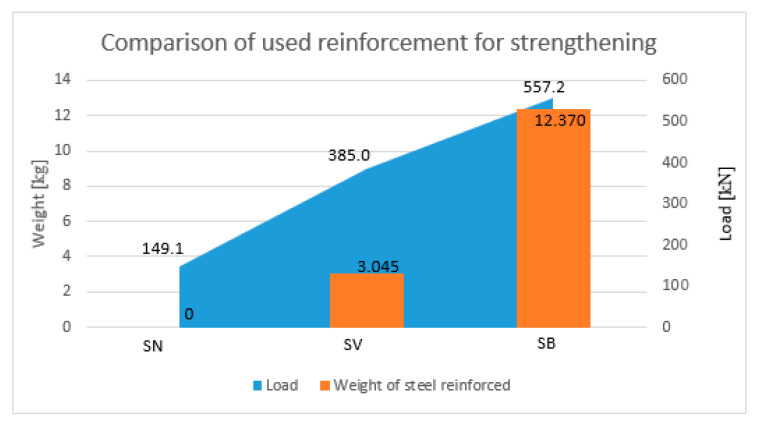
Comparison of used reinforcements for a single column reinforcement.

**Figure 15 materials-13-05432-f015:**
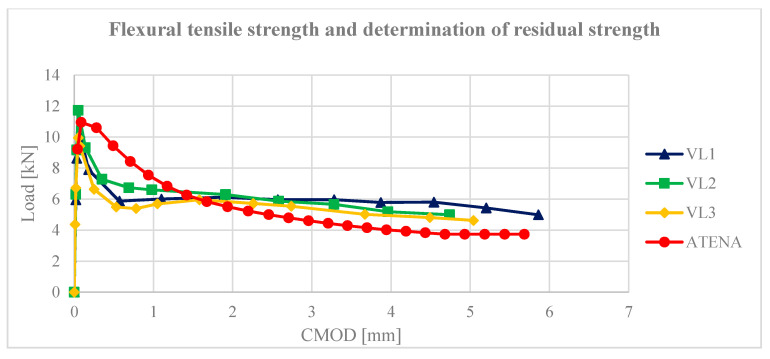
Flexural tensile strength—sample parameterisation.

**Figure 16 materials-13-05432-f016:**
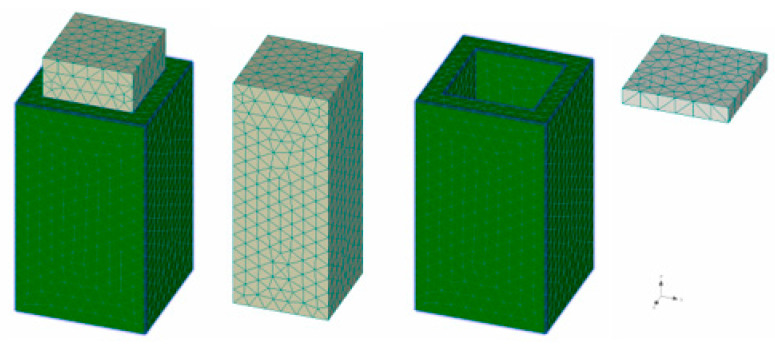
3D numerical model—Push test.

**Figure 17 materials-13-05432-f017:**
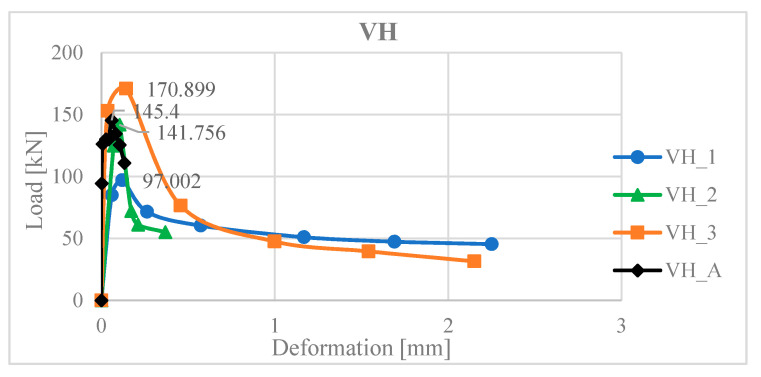
Graph of load-deformation dependence for the VH push test.

**Figure 18 materials-13-05432-f018:**
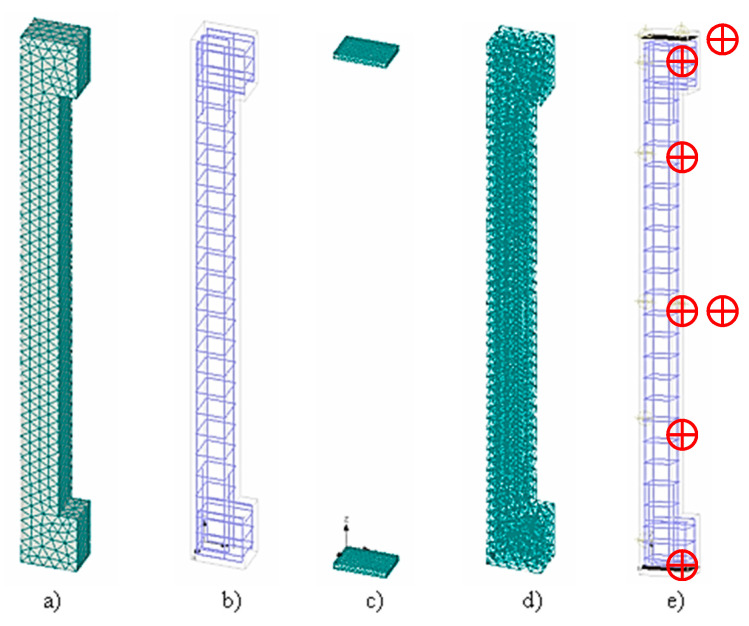
3D numerical model of the non-strengthened column, (**a**) 3D model, (**b**) reinforced bars, (**c**) steel plates, (**d**) mesh, (**e**) monitors point.

**Figure 19 materials-13-05432-f019:**
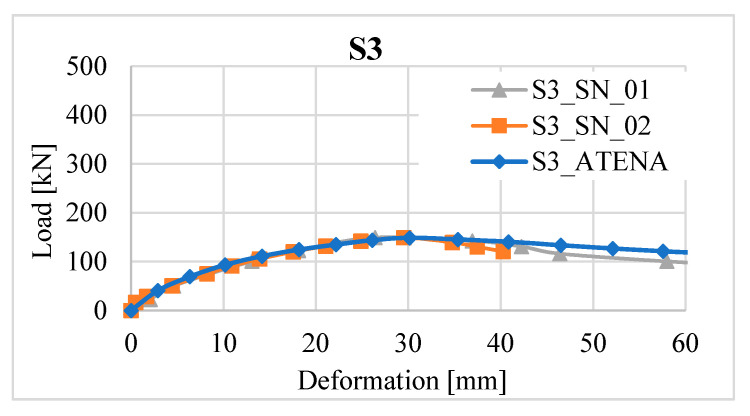
Load-deformation diagram for columns SN_01 and SN_02, numerical model.

**Figure 20 materials-13-05432-f020:**
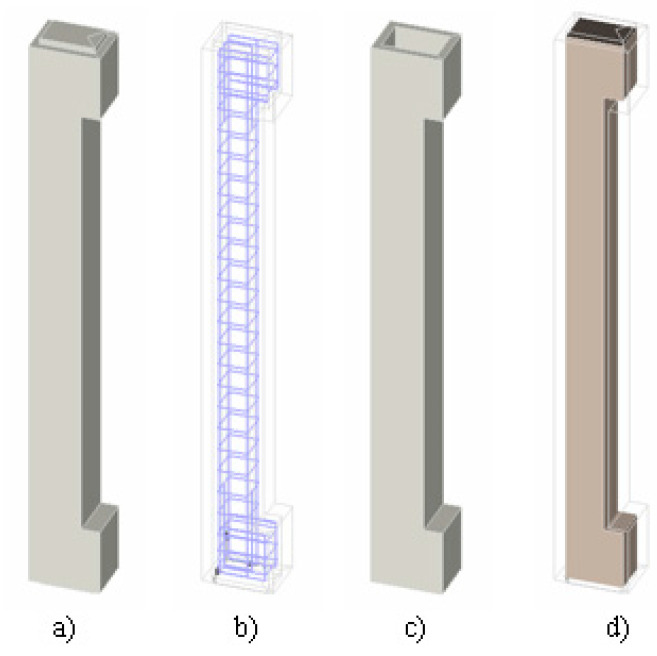
3D numerical model of strengthened column, (**a**) 3D full model, (**b**) reinforcement, (**c**) encasement (new layer of fibre reinforced concrete), (**d**) core (reinforced concrete).

**Figure 21 materials-13-05432-f021:**
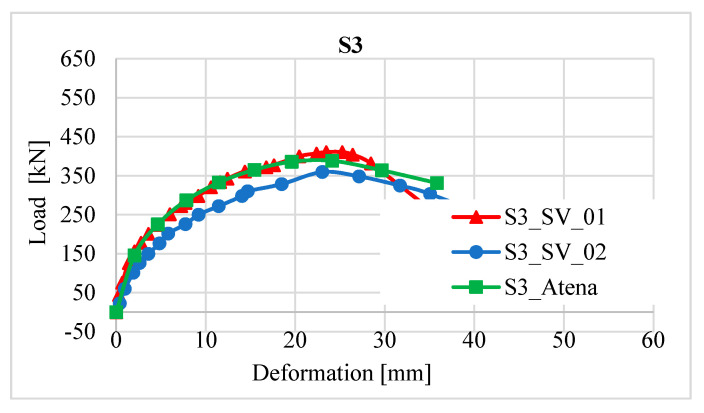
Load-deformation diagram for columns SV_01 and SV_02, numerical model.

**Figure 22 materials-13-05432-f022:**
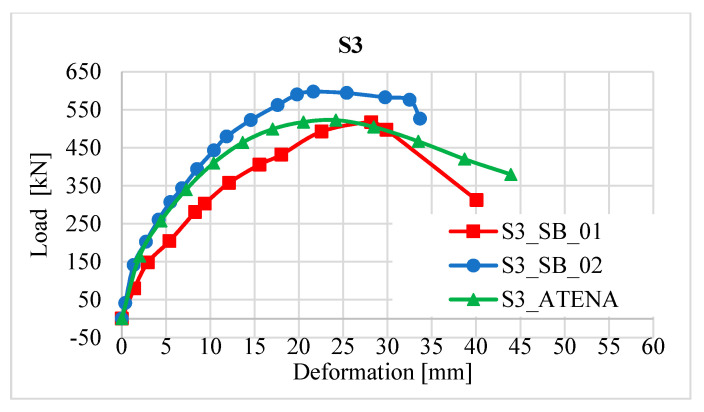
Load-deformation diagram for columns SB_01 and SB_02, numerical model.

**Table 1 materials-13-05432-t001:** Design of the concrete mixture.

Concrete Mixture	1 kg/m^3^
CEM II/B-S 32,5R (Ladce)	400
Aggregate 0/8 mm	910
Aggregate 8/16 mm	685
Fly ash (USS Košice)	80
Water	200
Fibres DRAMIX 3D	40

**Table 2 materials-13-05432-t002:** Cube strength of the fibre reinforced concrete (FRC).

Numbers	Bulk Density [kg m^−3^]	*f_c_* [kN]	*f_c,cube_* [MPa]
A1	2215.6	986.9	43.61
A2	2255.6	986.0	43.95
A3	2241.5	900.2	40.27
B1	2269.8	1162.2	52.27
B2	2253.8	1083.3	47.87
B3	2264.4	1079.5	47.99

**Table 3 materials-13-05432-t003:** Results from experiments—forces at sample failure F_max,failure_—“push tests”.

Mark	Measured Values *F_failure,exp_* (kN)	Average Value *F_failure,average_* (kN)	Average Calculated Value *F_failure,calc_* (kN)	Mark	Measured Values *F_failure,exp_* (kN)	Average Value *F_failure,average_* (kN)	Average Calculated Value *F_failure,calc_* (kN)
VH_1	97.002	136.552	38.4	FH_1	28.692	31.183	0.0
VH_2	141.756			FH_2	33.675		
VH_3	170.899			ZZ_1	228.806	262.171	96.0
PH_1	69.574	88.378	38.4	ZZ_2	298.120		
PH_2	116.71			ZZ_3	259.589		
PH_3	82.851			ZR_1	282.027	238.417	78.6
ZH_1	172.170	137.033	38.4	ZR_2	245.530		
ZH_2	93.191			ZR_3	187.709		
ZH_3	145.738						

**Table 4 materials-13-05432-t004:** Cohesion—comparison values from experiments and calculation.

Mark	Shear Resistance from Experiment *F_failure,average_* (kN)	Shear Resistance According to Standard *F_failure,calc_* (kN)	Cohesion from Reverse Calculating *c_reverse_* (MPa)	Cohesion According to Standard *c_calc_* (MPa)	Differences of Cohesion *c* (%)
VH	136.552	38.4	0.71	0.20	28.16%
PH	88.378	38.4	0.46	0.20	43.47%
ZH	137.033	38.4	0.71	0.20	28.16%
FH	31.183	-	-	-	-
ZZ	262.171	96.0	1.36	0.50	36.76%
ZR	238.417	78.6	1.24	0.40	32.25%

**Table 5 materials-13-05432-t005:** Maximum load forces of individual columns during the column failure.

Brand	Measured Failure Force (kN)	Maximum Deformation in Middle of Span, Sensor S3 (mm)
SN_01 *	149.512	26.39
SN_02 *	147.503	30.16
SV_01 *	410.622	25.23
SV_02 *	359.343	23.03
SB_01 *	597.416	21.63
SB_02 *	517.102	28.16

* SN—non-strengthened columns, SV—column strengthened with FRC layer (wrapping), SB—column strengthened with reinforced concrete layer (wrapping).

**Table 6 materials-13-05432-t006:** Numerical modified properties for fibre reinforced concrete.

3D Nonlinear Cementitious 2	
Property	Value
Elastic modulus *E* (MPa)	37 × 10^3^
Poisson’s ratio *μ* (-)	0.200
Tensile strength *F_t_* (MPa)	1,350
Compressive strength *F_c_* (MPa)	−39.50
Fracture energy *G_f_* (MN/m)	1.750 × 10^−4^
Critical compressive disp. *W_d_* (m)	−1.50 × 10^−3^
Plastic strain at strength *ε_cp_* (-)	−1.520 × 10^−3^

**Table 7 materials-13-05432-t007:** 3D interface contact—fibre reinforced concrete, smooth surface.

3D Interface		
Normal stiffness *K_nn_*	6.00 × 10^8^	MN/m^3^
Tangential stiffness *K_tt_*	2.75 × 10^7^	MN/m^3^
Tangential force *F_t_*	1.00 × 10^−1^	MPa
Cohesion *c*	0.35	MPa
Friction coefficient *µ*	0.5	-
Min. normal stiffness *K_nn,min_*	3.50 × 10^4^	MN/m^3^
Min. tangential stiffness *K_tt,min_*	1.00 × 10^6^	MN/m^3^

**Table 8 materials-13-05432-t008:** Properties of concrete in the numerical model.

3D Nonlinear Cementitious 2	
Property	Values
Elastic modulus *E* (MPa)	2.25 × 10^4^
Poisson’s ratio *μ* (-)	0.195
Tensile strength *F_t_* (MPa)	1.10
Compressive strength *F_c_* (MPa)	−18.50
Fracture energy *G_f_* (MN/m)	8.75 × 10^−5^
Critical compressive disp. *W_d_* (m)	−0.0025
Plastic strain at strength *ε_cp_* (-)	−1.050 × 10^−4^

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
