# Peer review of "Strengthening of Concrete Column by Using the Wrapper Layer of Fibre Reinforced Concrete"

_materials, 2020, doi:10.3390/ma13235432_

Round 1

Reviewer 1 Report

This paper provides the test results to show the friction(or bond) behavior between old reinforced concrete and new concrete with or without fibers. Since the bond between old and new concrete layers is quite important on the structural behavior of the member strengthened with a wrapping, I think this manuscript could be very useful in the relevant research areas. I put some review comments which might be helpful to enhance the manuscript, as follows;

1) line 14

'be increase of' -> 'be increased of'

2) Terminology; fiber concrete

Generally, 'fiber reinforced concrete (FRC)' is used, not 'fiber concrete'.

3) Figure 1 and the relevant test method to evaluate the tensile behavior of FRC

Many researchers conducted direct tension tests to evaluate the tensile behavior of FRC. It is recommended to add the direct tension test to Fig. 1 and the manuscript, with the consideration of the following references;

S-C Lee, J-H Oh, and J-Y Cho (2016) “Fiber Efficiency in SFRC Members Subjected to Uniaxial Tension,” Construction and Building Materials, Vol 113, pp. 479-487.

L. G. Sorelli, A. Meda, and G. A. Plizzari (2005) "Bending and Uniaxial Tensile Tests on Cocnrete Reinforced with Hybrid Steel Fibers," Journal of Materials in Civil Engineering, Vol. 17, No. 5, pp. 519-527.

4) line 121, 'minimal increase in cross-section'

In general, wrapping methods with FRP sheet or steel plate have minimal increase in cross-section, rather than fiber reinforced concrete. Please revise the sentence accordingly.

5) Figure 5

Although the notations for the specimens are provided on page 4, it is recommended to provide the notations in Figure 5 as well.

6) Equations (8)~(9), slip test method, and the relevant explanation in the manuscript with the test method

How did you evaluate the friction coefficient from the test results? Since there is no normal stress through the transverse direction in the test, in my opinion, the test results only indicated cohesion in Eq.(8).

7) Average calculated value in Table 3

How did you calculate the average calculated value in Table 3? Please provide details in the table, especially with the cohesion and friction coefficient used in the calculation.

8) Figure 11

Please provide details about the area with strengthening.

9) Notation in Table 4

Please provide an explanation of notation (SN, SV, and SB).

10) lines 319~328

It is hard to agree that the effect of strengthening is proportional to the weight of the material used for strengthening. For example, in fiber reinforced concrete, the tensile stress is not proportional to the fiber contents because of two reasons; 1) as fiber contents increase, the tensile behavior of FRC changes from tension softening behavior to strain-hardening behavior, 2) as fiber contents increase, fiber efficiency decreases. Please revise the paragraph accordingly.

11) line 400, 'calibrated'

What is the meaning of 'calibrated' here?

12) line 416

It is too simple. A detailed analysis is required.

13) Figure 21

Add the analysis results for the SB specimens.

Author Response

Dear Reviewer, thank you very much for your time in preparing the review and comments, which can improve the quality of our article.

We will answer you gradually after your points.

 1) line 14

'be increase of' -> 'be increased of'

Answer: We agree, it was corrected.

 2) Terminology; fiber concrete

Generally, 'fiber reinforced concrete (FRC)' is used, not 'fiber concrete'.

Answer: We agree, it was corrected.

 3) Figure 1 and the relevant test method to evaluate the tensile behavior of FRC

Many researchers conducted direct tension tests to evaluate the tensile behavior of FRC. It is recommended to add the direct tension test to Fig. 1 and the manuscript, with the consideration of the following references;

S-C Lee, J-H Oh, and J-Y Cho (2016) “Fiber Efficiency in SFRC Members Subjected to Uniaxial Tension,” Construction and Building Materials, Vol 113, pp. 479-487.

  1. G. Sorelli, A. Meda, and G. A. Plizzari (2005) "Bending and Uniaxial Tensile Tests on Cocnrete Reinforced with Hybrid Steel Fibers," Journal of Materials in Civil Engineering, Vol. 17, No. 5, pp. 519-527.

Answer: We agree. The figure with direct tensile and also literatures have been supplemented.

4) line 121, 'minimal increase in cross-section'

In general, wrapping methods with FRP sheet or steel plate have minimal increase in cross-section, rather than fiber reinforced concrete. Please revise the sentence accordingly.

Answer: We agree, the sentence has been revised.

 5) Figure 5

Although the notations for the specimens are provided on page 4, it is recommended to provide the notations in Figure 5 as well.

Answer: The notations of the specimens have been supplemented into figure 5.

 6) Equations (8)~(9), slip test method, and the relevant explanation in the manuscript with the test method

How did you evaluate the friction coefficient from the test results? Since there is no normal stress through the transverse direction in the test, in my opinion, the test results only indicated cohesion in Eq.(8).

Answer: We agree, the nominal stresses are equal to zero, so the friction coefficient is not needed to consider. It was revised – supplemented the sentences. We hope that now it is clear.

 7) Average calculated value in Table 3

How did you calculate the average calculated value in Table 3? Please provide details in the table, especially with the cohesion and friction coefficient used in the calculation.

Answer: The values of cohesion and friction coefficient were added into text with more details. We hope that now it is clear.

 8) Figure 11

Please provide details about the area with strengthening.

Answer: The areas have been filled into figure. We hope that we did it the way you thought.

 9) Notation in Table 4

Please provide an explanation of notation (SN, SV, and SB).

Answer: The explanations have been supplemented to the table.

 10) lines 319~328

It is hard to agree that the effect of strengthening is proportional to the weight of the material used for strengthening. For example, in fiber reinforced concrete, the tensile stress is not proportional to the fiber contents because of two reasons; 1) as fiber contents increase, the tensile behavior of FRC changes from tension softening behavior to strain-hardening behavior, 2) as fiber contents increase, fiber efficiency decreases. Please revise the paragraph accordingly.

Answer: We agree with you, but maybe it wasn't understood the way we thought it was. We agree that above about 1.5 - 2% of fibers, these fibers no longer bring the same effect as expected. A ratio of 0.5% was used in our samples. We just wanted to point out that per unit weight of steel we achieved better utilization efficiency than in the case of reinforced concrete. We have corrected and supplemented the sentences, so we hope that it is clearer now.

 11) line 400, 'calibrated'

What is the meaning of 'calibrated' here?

Answer: The world “calibrated” was changed to “adopted”. It means that we changed the parameters in the numerical model so that the results match the average values obtained from the experiment.

 12) line 416

It is too simple. A detailed analysis is required.

Answer: We agree, we have expanded the conclusions in which there is more information. We hope that now it is clear.

 13) Figure 21

Add the analysis results for the SB specimens.

Answer: The results for the SB specimens have been added.

Reviewer 2 Report

  • Alternative for the confinement of columns needs to be considered in order to make the background more complete. Please consider the following:
    • Cascardi, A., Aiello, M. A., & Triantafillou, T. (2017). Analysis-oriented model for concrete and masonry confined with fiber reinforced mortar. Materials and Structures, 50(4), 202.
    • Faleschini, F., Zanini, M. A., Hofer, L., & Pellegrino, C. (2020). Experimental behavior of reinforced concrete columns confined with carbon-FRCM composites. Construction and Building Materials, 243, 118296.
  • In lines 126-128 is stated that “The load-bearing capacity of the reinforced element (the whole cross-section) depends in part 126 on the shear resistance between the old concrete (core) and a layer of the newly added concrete”. This is not true in case of confinement which is a contact depended phenomena due to the lateral dilatation of the inner core. Please better comment this aspect.
  • In fig 9 please use the same scale within the axes.
  • Where is sensor 3 installed?
  • Parametric analysis of the numerical model is useful for the evaluation of the robustness of the simulation.
  • Conclusion section is too short. Please add more details.

Author Response

Dear Reviewer, thank you very much for your time in preparing the review and comments, which can improve the quality of our article.

We will answer you gradually after your points.

  • Alternative for the confinement of columns needs to be considered in order to make the background more complete. Please consider the following:
    • Cascardi, A., Aiello, M. A., & Triantafillou, T. (2017). Analysis-oriented model for concrete and masonry confined with fiber reinforced mortar. Materials and Structures, 50(4), 202.
    • Faleschini, F., Zanini, M. A., Hofer, L., & Pellegrino, C. (2020). Experimental behavior of reinforced concrete columns confined with carbon-FRCM composites. Construction and Building Materials, 243, 118296.

Answer: We agree that the background could be more complete, so the literature were included in the list of literature. Thank you.

  • In lines 126-128 is stated that “The load-bearing capacity of the reinforced element (the whole cross-section) depends in part 126 on the shear resistance between the old concrete (core) and a layer of the newly added concrete”. This is not true in case of confinement which is a contact depended phenomena due to the lateral dilatation of the inner core. Please better comment this aspect.

Answer: We agree. You are right. We tried to explain it better and supplement it with sentences.

  • In fig 9 please use the same scale within the axes.

Answer: We agree. The scale of the axis has been unified.

  • Where is sensor 3 installed?
  • Parametric analysis of the numerical model is useful for the evaluation of the robustness of the simulation.

Answer: We agree.

  • Conclusion section is too short. Please add more details.

Answer: We agree. The conclusion was extended. We hope that now it is better.

Reviewer 3 Report

A careful review of the manuscript “Strengthening of concrete column by using the wrapper layer of fiber concrete“ has been completed. Despite the fact that the authors discuss a layer of the fiber concrete for columns´ strengthening, it is not really clear what is the main objective of the manuscript and how these systems can be implemented in practice; also it is very difficult to follow what is presented. This paper is useful for engineers to improve the rehabilitation quality of damage structures. However, this investigation is not comprehensive and there are still rooms to improve. Therefore, this manuscript is not recommended for publication in International Journal of Materials due to the fact that paper has a critical and serious problem explained below:

  1. English needs to be improved. I had difficulty to follow the text and had to read the same sentence several times.
  2. The originality is not explained in detail.
  3. References are excessive and not related most of the time.
  4. Please provide more detailed definitions on the analysis of the shear connection used in this paper.

I recommend that the paper is rejected in its current form and a request is made to revise and re-submit for review.

Author Response

Dear Reviewer, thank you very much for the time spent in preparing the review, even though it is not positive.

We tried to supplement and modify the article according to your comments and comments from two other opponents/reviewers, which reviews were positive. We hope that we have succeeded in improving the contribution to such an extent that it is more comprehensible and accepted.

We will answer you gradually after your points.

  1. English needs to be improved. I had difficulty following the text and had to read the same sentence several times.

Answer: We did the best we knew, the English was checked and corrected by prof. Nikolic, who studied at MIT in the USA some time ago. If the Editor requires it, we can check it out with the translation company.

  1. The originality is not explained in detail.

Answer: We tried to better describe the meaning of our contribution and its originality in the introduction. The originality of the article is the use of FRC to strengthen the vertical elements such as columns (so far FRC has been used mainly to reinforce horizontal elements as beams) by concreting with FRC layer (wrapping), and investigating the influence of different types of surface treatment (contact) between layers (old and new). We haven't found articles and researches in this area that anyone would do like us.

  1. References are excessive and not related most of the time.

Answer: We tried to cite the relevant literature with respect to the ideas in the sentences and whole paper, on the recommendation of the other two opponents we added another 4 pieces of literature.

  1. Please provide more detailed definitions on the analysis of the shear connection used in this paper.

Answer: We have added texts (sentences), extended the conclusions, we hope that the analysis is now clearer and better described.

Round 2

Reviewer 1 Report

All questions are answered accordingly.

Reviewer 2 Report

The paper was improved and can be now accepted in the present form 

Reviewer 3 Report

I think this still needs some work!